# Graphene Oxide/Polyvinyl Alcohol–Formaldehyde Composite Loaded by Pb Ions: Structure and Electrochemical Performance

**DOI:** 10.3390/polym14112303

**Published:** 2022-06-06

**Authors:** Alaa Fahmy, Badawi Anis, Paulina Szymoniak, Korinna Altmann, Andreas Schönhals

**Affiliations:** 1Department of Chemistry, Faculty of Science, Al-Azhar University, Cairo 11884, Egypt; 2Bundesanstalt für Materialforschung und -Prüfung (BAM), Unter den Eichen 87, 12205 Berlin, Germany; paulina.szymoniak@bam.de (P.S.); korinna.altmann@bam.de (K.A.); 3National Research Centre, Spectroscopy Department, Physics Research Institute, 33 El Bohouth Street, Giza 12622, Egypt; badawi.ali@daad-alumni.de

**Keywords:** graphene oxide, polyvinyl formaldehyde, lead ions, dynamic mobility, conductivity

## Abstract

An immobilization of graphene oxide (GO) into a matrix of polyvinyl formaldehyde (PVF) foam as an eco-friendly, low cost, superior, and easily recovered sorbent of Pb ions from an aqueous solution is described. The relationships between the structure and electrochemical properties of PVF/GO composite with implanted Pb ions are discussed for the first time. The number of alcohol groups decreased by 41% and 63% for PVF/GO and the PVF/GO/Pb composite, respectively, compared to pure PVF. This means that chemical bonds are formed between the Pb ions and the PVF/GO composite based on the OH groups. This bond formation causes an increase in the T_g_ values attributed to the formation of a strong surface complexation between adjacent layers of PVF/GO composite. The conductivity increases by about 2.8 orders of magnitude compared to the values of the PVF/GO/Pb composite compared to the PVF. This means the presence of Pb ions is the main factor for enhancing the conductivity where the conduction mechanism is changed from ionic for PVF to electronic conduction for PVF/GO and PVF/GO/Pb.

## 1. Introduction

The further development of modern electronics requires eco-friendly multifunctional materials with adaptable dielectric materials having outstanding electrical/dielectric characteristics. Due to the rapid charging-discharging rates, dielectric capacitors are receiving much great attention in their implementation in recent energy storage systems [1]. Several niobate, titanate, and antiferroelectric-based ceramics materials have been used in the preparation of efficient energy storage materials. However, ceramic-based energy storage materials suffer from high brittleness possibility and low electrical breakdown strength.

The task of designing, for example, tools, especially for the storage of energy, requires materials with a high dielectric permittivity in addition to low dielectric losses. Owing to their properties such as ease of processability, high flexibility, low cost, high electric breakdown strength and low dielectric losses, filled polymer-based materials are suitable candidates for energy storage [2,3]. However, their dielectric constant is relatively lower as compared with ceramic-based dielectric materials. This drawback can be overcome by introducing an appropriate filler into the polymeric matrix. High concentrations of ceramic materials have been used as a filler to enhance dielectric polymers’ properties [4]. The overall dielectric properties were enhanced; however, the low compatibility between the ceramic materials and the polymeric matrix results in decreasing the flexibility of the polymer in addition to increasing the electrical losses [5,6]. Among other carbon fillers, graphene has revealed a high potential as a nanofiller for polymer nanocomposites. This is due to the high (specific) surface area values and the high mobility of the charge carriers. Furthermore, the carbon material has an ultra-high mechanical strength in combination with outstanding electrical as well as thermal conductivity [7]. Therefore, the engineering potential of such nanomaterials can lead to applications in fields such as microelectronics, energy storage devices, memory chips, electromagnetic interference (EMI) shielding, hydrogen storage systems and membranes [8,9,10,11,12]. Graphene oxide, denoted further as GO, is a suited nanofiller for the preparation of graphene-containing polymer composite. As a pseudo-two-dimensional material, it has extraordinary mechanical [13], optoelectronic [14], and transport properties [15]. Moreover, GO as a material itself has numerous applications as a photocatalyst, supercapacitor, drug delivery system, and material for memory devices as well as for optoelectronics [16,17]. Moreover, GO carries a high number of oxygen-containing functional groups, a large surface area, and is well interconnected and porous, specifying plenty of oxygenated functional groups available for metal adsorption [18,19,20]. Furthermore, it demonstrates efficient antibacterial activity [21]. To combine the properties of GO with those of polymers it would be attractive to obtain an environmental polymer/GO composite material for advanced scientific applications. 

Poly(vinyl alcohol–formaldehyde) (PVF) has a sponge-like structure. It is a hydrophilic biocompatible polymer with excellent mechanical properties and an open-cell structure leading to a low density in addition to an outstanding water absorption capacity [22]. The ratio of the vinyl alcohol (VA) units in poly(vinyl acetal) is higher than 13.5% [23]. This provides the possibility to functionalize PVF because of the high number of hydroxyl groups in its structure [22].

Besides the applications discussed above, several carbon-based nanomaterials, such as graphene, carbon nanotubes (CNTs), and GO, have been developed as effective inorganic and organic pollutant adsorbents [24]. Jeyakumar et al. discussed the adsorption of lead (II) ions as adsorbates. The activated carbon material was made from (marine) green *Ulva fasciata* sp. CAC, as a commercially activated carbon, was used as an adsorbent under various conditions, such as varying pH values and contact times, in addition to varying concentrations of the adsorbate and activated carbon concentrations. The authors found that the reaction kinetics are of a pseudo-second-order and *Ulva fasciata* sp. is a suitable material for abstracting Pb ions out of aqueous solutions [25]. In difference polymer/GO composites, even GO cannot act only as an adsorbent for heavy metals but also for organic solvent [22] and gases, such as CO_2_ [26]. It might be that adsorbed species, like Pb ions, will enhance the electrical properties of the composites, which makes them even more suited for applications in sensors, etc.

Most researchers have focused on the kinetics of the removal process of Pb ions [27] in dependence on the pH value of the solution [28], the contact time, and the concentration of heavy metals [29], in an isotherm study [30] and how such a material can be applied as an adsorbent for metal ions. In our previous work, we studied Pb removal from aqueous media using a PVF/GO composite. Factors affecting the adsorption capacity, pH values, initial adsorbent dosage, initial metal ion concentration, contact time, regeneration, and isotherm were discussed [31]. However, the relationship between the structure and the electrochemical behavior of the composite film related to the absorbed metal ions has currently not been studied. It is well known that the adsorbents could be regenerated several times, and after that, they will be waste. Therefore, this research paper is focused on the structure and electrochemical properties of the composite film, which acts as an adsorbent to find new properties which may lead to new applications. Thus, here, dielectric spectroscopy is utilised as a tool to explore the segmental fluctuations as a molecular probe for the structure of the prepared PVF/GO composite in comparison to the corresponding composites loaded with Pb ions. Additionally, this paper also intends to explore the effect of the influence of the Pb ions on the structural and morphological properties as well as the dielectric behavior, including the electrical conductivity of the PVF/GO composites.

## 2. Materials and Methods

### 2.1. Materials

Expandable graphite flakes with a nominal lateral size of 300 µm were purchased from Asbury Carbons, USA. Polyvinyl alcohol (PVA) with a hydrolysis degree of 98–99% was obtained from LOBA Chemie. Triton X-100, potassium permanganate (KMnO_4_) and analytical-grade lead nitrate were acquired from Merck. Formaldehyde (38 wt%), hydrochloric acid (HCl, 37%), hydrogen peroxide (H_2_O_2_, 35%), sulfuric acid (H_2_SO_4_, 98%), and hypo-phosphoric acid (H_2_PO_4_, 85%) were provided by Fisher.

### 2.2. Methods

#### 2.2.1. Graphene Oxide

The improved method of Hammers was used to prepare monolayers of highly oxidized graphene oxide (GO) [32]. The dry graphite flakes were fast-heated in a tube furnace to 1323 K (1050 °C) for 50 s to obtain thermally expanded graphite (TEG).

The TEG (3 g) was dispersed in a mixture of H_2_SO_4_/H_2_PO_4_ (9:1) with continuous stirring overnight. The acid-graphite solution was pre-cooled to 268 K (−5 °C) in an ice bath, and then 18 g of KMnO_4_ were carefully added under stirring. The color of the acid-graphite solution became greenish-black during this process. After removing the ice bath, the mixture was heated to 323 K (50 °C) for 3 h and stirred for 72 h continuously at ambient conditions. The solution became light brown in color with a paste-like structure. Then the obtained graphite product was dissolved in small amounts of 400 mL of deionized ice water. After that, 7 mL of 30% hydrogen peroxide was slowly added to the graphite oxide dispersion (Figure 1).

After a few minutes, 400 mL of distilled water was added to the graphite oxide mixture. Then, the mixture was left for a few hours to allow the graphite oxide to precipitate. The precipitate was filtered. After that, it was washed several times with a 10 wt% HCl solution. Then it was washed with deionized water (3 times), ethanol (3 times) and deionized water (3 times) with centrifugation as well as decantation until a pH value of nearly 5 was reached. The precipitate was dried in a vacuum oven at 333 K (60 °C) overnight. The prepared GO was investigated elsewhere [31].

#### 2.2.2. Pure PVF Foam

The preparation of PVF was given in Figure 1. A 10 wt% solution of PVA was obtained by adding 50 g PVA to 450 mL distilled water at a temperature of 368 K (95 °C) for 180 min. Under stirring, formaldehyde (10 mL) was added to 60 g of the PVA solution. Moreover, 1 mL of 1% *v*/*v* Triton X-100 solution was added under hard stirring for 10 min, then cooled down to room temperature. After that cooling process, 20 wt% of H_2_SO_4_ (30 mL) was added to the solution under hard stirring for 10 min.

The foaming process of the PVA/formaldehyde solution was completed by annealing the solution at 60 °C for 5 h. The formed PVF foam was removed from the furnace, washed with hot water several times, and then dried at 60 °C for 2 h.

#### 2.2.3. GO/PVF Foam

GO powder (200 mg) was dispersed in distilled water (30 mL) using a tip-sonicator for 120 min where the temperature was controlled by an ice bath. To remove un-exfoliated GO flakes, the GO solution was then centrifuged at 5000 rpm for 60 min. Then, the GO solution was mixed with 60 g of a hot PVA solution (10 wt%) under stirring for 30 min. The foam was then obtained for PVF. The GO content in the material was 0.2 wt%. For more details, see Ref. [31].

#### 2.2.4. GO/PVF/Pb Foam

The GO/PVF composite was sliced into small pieces for the adsorption of Pb(II) ions on the surface of the film; the foam was sliced and divided into small pieces. Hot water was then used to clean the GO/PVF pieces several times to remove any impurities. The foam pieces were dried. Before the adsorption experiment, the samples were weighed. The adsorption capacity was defined by Q_t_ = V(C_0_ − C_e_)/W, where W (g) is the weight of the adsorbent, V is the volume of solution, and C_e_ and C_0_ (mg/L) are the initial and final concentration of solutions. The Q_t_ of GO/PVF foam as an eco-friendly and superior sorbent for Pb ions was studied in detail in our recent work based on pH value, initial adsorbent dosage, the given metal ion concentration and the contact time with the GO/PVF foam [31]. The optimum conditions to remove more than 99% of the Pb ions were: GO/PVF foam pieces (0.4 g) were washed with distilled water 10 times. The foam pieces were immersed in 100 mL of Pb ion solution with a concentration of 200 ppm. The solution was kept at room temperature under shaking (200 rpm) for 90 min. The pH value was adjusted to 5 during the experiment to avoid the formation of soluble hydroxyl complexes [28,29]. The foam pieces were removed from the solution of the Pb ions and washed with distilled water to eliminate any unabsorbed Pb(II) which might still be present on the surface of the sample. Then the GO/PVF/Pb pieces were dried under vacuum at 333 K (60 °C) for 120 min.

The adsorption process of lead ions on the PVF/GO sample is often modeled by a kinetic model of pseudo-second-order with a correlation coefficient (R^2^) of 0.989 [31]. The theoretical value of the amount of adsorbed ions at equilibrium q_e_ (mg g^−1^) is in good agreement with the estimated value [31]. Therefore, it can be stated that the pseudo-second-order model is the most suitable kinetic model describing the mechanism for the adsorption of Pb(II) onto the prepared foam.

The results have also revealed that the uptake process is also in good agreement with the isotherm model of Freundlich [33] with an R^2^ value of 0.9973 [31]. These results display that the adsorption of Pb ions took place on a heterogeneous GO/PVF surface. Moreover, in the frame of the adsorption isotherm model of the Dubinin–Kaganer–Radushkevich (DKR) adsorption isotherm model [34], an activation energy E_a_ of 10.7 kJ/mol (8 < E_a_< 16 kJ/mol) was obtained, indicating that the mechanism of adsorption is an ion exchange process [35].

### 2.3. Characterization Methods

#### 2.3.1. X-ray Photoelectron Spectroscopy (XPS)

The nature and concentration of the functional groups at the surface (information depth of 5 nm to 7 nm) of the PVF/GO/Pb composites compared to PVF/GO were investigated using XPS by considering the C1s and O1s peaks. A SAGE 150 spectrometer (Specs, Berlin, Germany) was utilized for these investigations. It used a Phoibos 100 MCD-5 hemispherical analyzer. AlKα X-ray radiation (non-monochromatic) was applied. The pressure in the analysis chamber was set to ca. 1 × 10^−7^ Pa. The angle between the axis of the analyzer lenses and the X-ray source was 54.9°, and the analyzer was fixed to an angle of 18° with respect to the surface normal. The constant analyzer energy (CAE) mode was used to record the XPS spectra. The examined area was ca. 1 × 3 mm^2^. The software routine of the manufacturer was applied for analyzing the XPS spectra in a quantitative way. Moreover, a computer-based surface analysis of XPS (CASA XPS) was utilized to fit the C1s spectra and to calculate concentrations of the functional groups. It was given in terms of 100 C atoms. All C1s spectra are referred to as the C-C peak, which is attributed to 284.6 eV for neutral carbon.

#### 2.3.2. FTIR Spectroscopy

The FTIR spectra were measured by a Nicolet Nexus 8700 FTIR spectrometer (Thermo Fisher Scientific, Waltham, WA, USA) and analyzed in the wavenumber range from 500 to 4000 cm^−1^. A total of 64 scans were accumulated with a resolution of 4 cm^−1^. A Diamond Golden Gate 1 reflection ATR accessory was equipped (Thermo Fisher Scientific, Waltham, MA, USA). The OMNIC software was used for the assessment of the peaks. Gaussians were fitted to the data. In contrast to XPS, the information depth of FTIR-ATR is up to 2.5 μm.

#### 2.3.3. Scanning Electron Microscopy

High-resolution scanning electron microscopy (HRSEM) utilizing a ZEISS Sigma 500 VP instrument was used to study the morphology of the foams. Energy dispersive X-ray (EDX) measurements were achieved in parallel by employing the EDX system included in the electron microscope.

#### 2.3.4. Differential Scanning Calorimetry (DSC)

DSC experiments were performed employing a power-compensated Perkin Elmer DSC 8500 device. The samples (ca. 3 mg) were measured using 50 μL aluminum pans in a wide range of temperatures from 243 K (−30 °C) to 463 K (190 °C), where the employed rates (cooling and heating) were 10 K min^−1^. The 2^nd^ heating runs were used for analysis. N_2_ gas (flow rate 0.333 mL/s) was employed as a purging agent. The baseline was estimated by recording an empty 50 μL aluminum pan under the same conditions as for the sample. The achieved baseline was subtracted from the heat flow measured for the sample. Moreover, indium was used for temperature calibration.

#### 2.3.5. Broadband Dielectric Spectroscopy (BDS)

The dielectric properties of the composites were measured by a high-resolution device based on an ALPHA analyzer (Novocontrol, Waltham, Germany) connected to a sample cell with an active head. The sample temperature was controlled by a Quatro device. The temperature deviation from the set value was smaller than 0.1 K. For further experimental specifics, see reference [36]. The geometry of parallel plates was applied for the measurements. The samples were mounted between two electrodes (material gold-plated brass, diameter 3 mm spacing 50 μm). Isothermal scans (frequency ranging from 10^−1^ Hz to 10^6^ Hz) were employed to measure the complex dielectric permittivity
*ε** (*f*) = *ε′*(*f*) − *iε″*(*f*) (1)
in the temperature range from 173 K to 473 K. *f* is the frequency where *ε′* and *ε″* are corresponding real and lost (imaginary) parts of the complex dielectric permittivity, where *i* = $\sqrt{-1}$ denotes the imaginary unit.

## 3. Results

### 3.1. Elemental Composition

XPS was employed to investigate the chemical compositions at the surface. The differences in the structure of PVF/GO and PVF/GO/Pb in comparison to pure PVF are shown in Figure 2 [30]. As expected, the C1s and O1s peaks were observed in all samples, which corresponds to carbon and oxygen groups, respectively. The composite PVF/GO has a similar oxygen content to the pure PVF sample. However, for PVF/GO/Pb, the Pb 4f peaks were observed, which corresponds to the adsorbed lead ions. It should be noted that the intensity of the O1s peak is increased in comparison to that of the C1s in the sample loaded with Pb ions, as observed in the XPS spectra (Figure 2) and Table 1.

The C1s XPS spectra for PVF material (see Appendix A) were separated into three contributions related to carbon atoms having different oxidation states due to the following functional groups containing oxygen: (1) nonoxygenated C-C bonds at a binding energy of 284.6 eV, (2) C-O bonds (binding energy of 286.3 eV) and (3) C=O bonds (binding energy of 287.5 eV) [37].

An additional component was observed at 288.3 eV for the PVF/GO and PVF/GO/Pb composites, which is assigned to O-C=O groups (Appendix A). This peak might also be related to the carboxylic and/or carboxylate groups [38]. For the PVF/GO, the amount of alcohol groups is decreased by 41%, whereas for the PVF/GO/Pb composite, it is reduced by 63% compared to pure PVF (Figure 3a). Hydrogen bonds and/or condensation processes between the OH groups in the PVF as well as OH or/and COOH groups of GO might be the reason for this reduction in the free OH groups for the PVF/GO composite.

This means that a chemical bond is formed between Pb ions and the PVF/GO composite based on the OH groups. However, the amount of O-C=O groups on the surface is increased ca. 35% after the adsorption process, as shown in Figure 3a. It is further revealed that the total amount of functional groups containing oxygen is increased, as shown in Table 1. This is probably due to the high amount of alcohol and carboxylic groups compared to other functional groups also bearing oxygen at the surface, enhancing the uptake of lead ions as discussed above.

The Pb4f XPS spectra for the PVF/GO/Pb composite (see Figure 3b) is deconvoluted into four components, at least, including Pb4f_7/2_ (metal), Pb4f_7/2_ (PbO_2_) Pb4f_5/2_ (metal) Pb4f_5/2_ (PbO_2_) at 136.9, 138.4, 141 and 143.2 eV, respectively. It was found that the sample consists of Pb metal and Pb oxide. Nevertheless, the main component is PbO_2,_ as shown in Figure 3b. Therefore, the total amount of oxygen-containing groups is increased after the adsorption process. Thus, the exchange of ions and/or the formation of complexes is the main mechanism for the adsorption of lead ions onto carbon adsorbents in an acidic medium. However, in an aqueous solution, the π electron systems of the graphene layers, which are delocalized, act as Lewis bases. This leads to the formation of electron donor-acceptor complexes between the H_2_O molecules and the lead ions [39]. The essential reaction from a Lewis acid A to a Lewis base B is mainly the formation of an A-B complex (addition or coordination complex). However, it is well known that acids act as electron-pair acceptors, whereas bases are electron-pair donors. For the Lewis acid-base reaction, the unshared electron pair of the base plays an important role in creating a coordination bond with an electron-missing atom of the acid [40]. For the composite and the Pb ions, the Lewis base is PVF/GO, while the metal ion is the Lewis acid. A complex formation among PVF/GO and Pb ions might take place via a Lewis A-B interaction. Consequently, it can be deduced that lead ions concurrently adsorbed onto PVF/GO composite, developing in an enlarge of the pH value [30].

### 3.2. FTIR Spectroscopy

FTIR was utilized not only to verify the structure of the prepared PVF and PVF/GO composite but also to study a possible interaction among the functional groups of PVF/GO and Pb ions in the whole sample.

The FTIR spectra of the PVF polymer, PVF/GO and PVF/GO/Pb are depicted in Figure 4. The FTIR spectrum of pure PVF reveals the following characteristic bands: at ≈3400 cm^−1^, the ν(OH) stretching vibration associated with ν(CH_3_^as^) is observed at ≈2940 cm^−1^, ν(CH_2_^as^) vibration is displayed at ≈2920 cm^−1^, ν(CH_3_^s^) is seen at ≈2890 cm^−1^, and the ν(CH_2_^s^) vibrations is observed at ≈2880 cm^−1^ with the related ν(C−O) vibration which appears at ≈1047 cm^−1^ [41]. The FTIR spectra exhibit the carbonyl bonds (like XPS) evidenced by the stretching vibration of C=O near 1700 cm^−1^. The vibrations in the range from 1500 cm^−1^ to 1300 cm^−1^ are assigned to the C-H bending [42]. The FTIR bands in the range from 1278 cm^−1^ to 1091 cm^−1^ are ascribed to stretching vibrations of the C-O-C ether linkages as well as of the O-H alcoholic groups [43]. The peak at 840 cm^−1^ is assigned to the vibration of a C-C group (stretching vibration). The band observed at 947 cm^−1^ is characteristic of the C-OH group of PVF [44,45].

The FTIR spectra of the PVF/GO and the PVF/GO/Pb composite exhibit several characteristic peaks such as those observed in the PVF polymer, as given in Figure 4. The area under the peaks of the stretching vibration for the free O-H and/or hydrogen bonds (O···H) (3600 to 3100 cm^−1^) are increased for PVF/GO compared to PVF related to OH and COOH groups in GO, as expected. However, the adsorption process of Pb ions onto the surface of the PVF/GO composite decreases the area of this peak. This decrease confirms the results obtained by the XPS measurement (the inset of Figure 4). The band at ≈1730 cm^−1^ is ascribed to the C=O stretching vibration of the O-C=O group [11,46,47]. The bands at ≈1025, 1255, as well as 1396 cm^−1^ are assigned to the stretching vibration of the C-O bond of the alkoxy, carboxylic and epoxy groups of GO.

Appendix A gives the stretching vibration of the C=O group in the wavenumber region between 1800 and 1500 cm^−1^ for PVF, PVF/GO and the PVF/GO/Pb composite. A detailed assessment shows that the assignment of this peak is extremely complex. It consists of three different contributions or even more.

To explore the role of the functional groups containing stretching vibration, the three Gaussians were fitted to the data for C=O groups. The component appearing at wave numbers around 1600 cm^−1^ is associated with the vibrations of the C=C group. The second one with a shoulder at 1640 cm^−1^ is connected to intermolecular hydrogen bonding. The contribution at wave numbers of around 1724 cm^−1^ is attributed to the C=O carbonyl stretching vibration of the O-C=O groups [48].

The groups of the PVF/GO composite containing oxygen react via complex surface complexation with the Pb ions. Therefore, intermolecular hydrogen bonding is reduced, which increases the intensity of the contribution of the O-C=O groups for the PVF/GO/Pb composite (Figure 5), confirming the results obtained from XPS measurement [49,50].

### 3.3. Surface Topography and Elemental Composition

High-resolution scanning electron microscopy (HRSEM) images of PVF, PVF/GO and the PVF/GO/Pb composite are provided in Figure 6. The PVF foam displays a sponge-like structure with an open-cell, interconnected, highly porous network (Figure 6a). The average pore size is ca. 6–8 μm. The surface roughness and porosity changed when GO was added to the polymeric matrix, as shown in Figure 6b. The GO sheets are supposed to be distributed homogeneously inside the PVF matrix during the acetalization and the foaming of PVA. This process allows the surface area of the GO sheets to be as large as possible. The super-hydrophilic properties of the sheets of GO in an aqueous solution are important for its homogenous distribution in the dissolved PVA chains. An aggregation of lead crystals on the surface of the PVF/GO composite was observed after the adsorption experiment, as shown in Figure 6c. The formation of these crystals was approved by EDX analysis (see inset of Figure 6d). The amount of adsorbed Pb is found to be ca. 4.33%. The atomic percent of Pb ions on the PVF/GO composite was investigated by EDX analysis after the adsorption process of the Pb ions (Figure 6d).

It was found that the Pb crystals are homogeneously dispersed on the surface of the PVF/GO material. Such behavior might have resulted from the homogenous dispersion of the GO sheets within the PVF matrix.

### 3.4. Glass Transition

The analysis of the DSC heat flow curves was carried out by the following procedure. A sigmoidal function was fitted to the heat flow data. The first derivative of the sigmoidal fit with respect to temperature was further taken, leading to a peak. The glass transition temperature T_g_ was estimated from the maximum position of this peak T_g_ (see Figure 7a–c). This strategy enables a consistent and unambiguous determination of T_g_ and minimizes the scattering of derivative data points. The obtained T_g_ values are given in Figure 7d.

T_g_ increases from pure PVF to the prepared PVO/GO composites from 111.2 °C to 127.5 °C. This means that the thermal stability of the PVF/GO foam is enhanced by the addition of GO. The chemisorption of Pb in the case of PVF/GO/Pb increases T_g_ further to 158.9 °C. This increase in T_g_ value could be attributed to the formation of strong surface complexation between adjacent layers in the PVF/GO/Pb composite. There is also a significant broadening of the glass transition region. The broadening is symmetric, indicating that the composites have some higher and lower mobility compared to the average [29]. The broadening suggests that the composite has a broader distribution of the glass transition temperatures, which means that there is some heterogeneity in the system induced by GO and even more by Pb.

### 3.5. Dielectric and Conductivity Measurements

#### 3.5.1. Molecular Mobility

The dielectric behavior of PVF/GO/Pb materials was investigated to explore their molecular mobility and conductivity and, therefore, their feasibility for energy storage applications. Appendix A provides the imaginary *ε″* of PVF (a) in a 3D representation as a function of frequency and temperature in a 3D representation for the heating cycle compared to the PVF/GO (b) and the PVF/GO/Pb composites (c). These 3D representations reveal that the dielectric spectra seem to be quite complex for all materials. Therefore, the dielectric loss is plotted in the isochronal representation as a function of temperature for selected frequencies in Figure 8. First pure PVF is considered (see Figure 8a).

At low temperatures, a peak is detected, which shifts with increasing frequency to higher temperatures. Such behavior indicates a relaxation process due to fluctuations of molecular groups. This process is assigned to localized fluctuations and denoted as β-relaxation. Unfortunately, the data have a considerable amount of scattering, which prevents a reliable analysis of the β-relaxation.

At higher temperatures, a second dielectric active process is found, displaying quite unusual behaviour. The maximum temperature of that process seems to be independent of frequency, and its dielectric strength increases strongly with decreasing frequency. Both properties of the process indicate that this mode cannot be assigned to a molecular relaxation process related to dipolar fluctuations. That it is observed in the temperature range where also DSC shows the glass transition might be accidental. In the literature, it was argued on the one side that such specific features are related to a percolation phenomenon [51]. On the other hand, in another study on porous silica glasses, such a behaviour was assigned to a percolation of electric excitation through the interconnected pore network [52,53,54]. Moreover, a similar process was observed in a polymer-based nanocomposite with layered double hydroxides as nanofiller [55]. As shown by electron microscopy, the PVF also has a porous, foam-like structure. Therefore, this peak might also be assigned to a percolation of electrical excitation through the interconnected pore network as discussed above. The further increase in the dielectric loss with increasing temperature and decreasing frequency is related to the conductivity related to the drift motion of charge carriers.

Figure 8b gives the dielectric loss versus temperature for different frequencies for the sample PVF/GO. The observed behavior is even more complex than that of pure PVF. For temperatures between 273 K (0 °C) and 323 K (50 °C) in comparison with the spectra of PVF, a further process is observed. It shifts to higher temperatures with increasing frequency. At first glance, it might be assigned to the β-relaxation of PVF. One must consider that the relaxation process observed for PVF/GO takes place at higher temperatures than the β-relaxation in pure PVF. It might be that the observed process has a similar molecular origin to the β-relaxation of pure PVF, but which is strongly modified by the GO nanoparticles. Nevertheless, from the literature, it is known that the localized character of the molecular fluctuations responsible for the β-relaxation is only less affected by the nanoparticles. It is, therefore, reasoned that the observed relaxation process for PVF/GO has a different molecular reason than the β-process for pure PVF. The process is further analysed in the temperature domain, and the maximum temperature is taken and plotted with the corresponding frequency in the Arrhenius plot (see Figure 8d). When plotted over 1/T, the data are linear and can be defined by the Arrhenius dependence defined by
(2)$$f_{p,\beta }\;=\;f_{\infty }\;\exp(-\frac{E_{a,\beta }}{k_{B}\;T})$$

Here, *f*_∞_ is the pre-exponential factor, *k_B_* is the Boltzmann constant, and *E_a_* denotes the activation energy.

The estimated activation energy of ca. 50 kJ/mol is rather high and not characteristic for a localized process. The process might be therefore related to molecular fluctuations taking place in interfacial regions between the matrix polymer and GO.

At higher temperatures, two processes are observed, which show similar behavior to the percolation process observed for PVF. Therefore, for PVF/GO, they are also assigned to a percolation of electrical excitations. That two processes are observed might point to a structural heterogeneity that is introduced by the nanoparticles. This agrees with the DSC measurements where a broadened glass transition zone is observed compared to pure PVF. Surprisingly no increase in the dielectric loss with increasing temperature is observed at high temperatures like for PVF, pointing to a conductivity contribution. This will be discussed in more detail below.

Figure 8c gives the dielectric spectra for PVF/GO/Pb versus temperature for different frequencies. At low temperatures, a relaxation process is observed, where its position shifts with increasing frequency to higher temperatures. The process is analysed in the same way as the process observed for PVF/GO. The data are given in addition to PVF/GO in Figure 8b. For PVF/GO/Pb, the process is monitored at lower temperatures than observed for PVF/GO but has a higher activation energy with a value of 79 kJ/mol. In this case, the Pb ions act as a filler which might be bonded to the polymer layers via OH and/or COOH groups forming a cross-linked structure in agreement with the data obtained from XPS, FTIR and DSC measurements. Therefore, it is concluded that hindering the fluctuation leads to an increase in *E_a_*. A further process is observed at even higher temperatures, which has the properties that the processes observed for PVF and PVF/GO. Here the process is quite broad, indicating a pronounced heterogeneity of the sample, which agrees with the DSC results.

#### 3.5.2. Conductivity

A conductivity contribution is observed for PVF, PVF/GO and PVF/GO/Pb composites in addition to the processes discussed above.

The complex dielectric permittivity *ε** and the complex conductivity *σ** are related by [56]:(3)$$\sigma^{*}\left(\omega \right)=\sigma^{\prime }\left(\omega \right)+i\sigma^{\prime \prime }\left(\omega \right)=i\omega \varepsilon_{0}\varepsilon^{*}\left(\omega \right).$$

*σ*′ and *σ*″ are the real and imaginary part of *σ**, defined by:(4)$$\sigma^{\prime }\left(\omega \right)=\omega \varepsilon_{0}\varepsilon^{\prime \prime }\left(\omega \right)$$

The frequency dependence of the real part of the complex conductivity *σ*′ is presented in Appendix A. It displays the typical frequency dependence of disordered materials [57]. Starting from high frequencies, *σ*′ decreases with decreasing frequency at high frequencies, obeying a power law until a plateau is reached at a characteristic frequency f_c_. f_c_ characterizes the onset of the dispersion, while the plateau value is related to the DC conductivity [56].

Figure 9a compares the real part of the conductivity for the different materials at the same temperature. This figure shows that the conductivity of PVF/GO/Pb is ca. three orders of magnitude higher than the other two materials.

In the literature, several models exist to explain the frequency dependence of s’(f), such as the Dyre model [58].

As an approximation, the conductivity spectra can also be described by the power-law introduced by Jonscher. By fitting the Jonscher equation to s’(f), the *DC* conductivity σ_DC_ is estimated [59]. The temperature dependence of σ_DC_(T) temperature dependence is depicted in Figure 9b.

For pure PVF, the DC conductivity increases with increasing temperature, as expected for a polymer, whereas conduction mechanism hopping processes are assumed to be triggered by the segmental fluctuations related to the glass transition. However, for the PVF/GO composites (Appendix A), the σ_DC_ values decrease with increasing temperature. This change in σ_DC_(T) points to a change in the mechanism of the conduction process. Reconsidering that GO, due to its chemical structure, will allow for electronic conductivity, one might conclude that the conductivity in PVF/GO is due to electrons in percolated GO sheets.

For PVF/GO/Pb, the dependence of the conductivity on temperature is even more complex (see Appendix A). For temperatures lower than 181 °C, the DC conductivity is approximately independent of temperature. In that temperature range, σ_DC_ is significantly higher than that for PVF and PVF/GO. It seems to be logical that the enhanced conductivity is due to the presence of the Pb ions. Surprisingly, for temperatures above 181 °C, the σ_DC_ decreases by approximately 2.3 orders of magnitude down to values observed for the PVF (see Figure 9b) [56]. This indicates a further change in the underlying conduction mechanism and, therefore, in the structure of PVF/GO/Pb. It might also indicate that the temperature is important for the structure of the PVF/GO/Pb composite. The change in the temperature dependence is not sharp but gradual, ruling out a phase transition. A complex formation between PVF/GO and the Pb ions can occur through Lewis acid-base interactions, as discussed in the XPS section. It might be that this complex is stable at low temperatures, leading to a high but temperature-independent DC conductivity. At higher temperatures, this might become unstable also due to the increased segmental mobility leading to conventional electronic conductivity. Another possibility is that the Pb ions will be reduced at higher temperatures and lose their high conductivity [60]. Nevertheless, these effects need further investigation.

## 4. Conclusions

The new production of materials is being requested, such as solid-state electrolytes that are more effective for ion conduction in mechanically robust yet cheap polymer materials. Here, GO/PVF composites are synthesized that have high chemical stability. The developed sorbent foam is found to be a superb adsorbent material for the removal of Pb(II) with an excellent reusability of more than 10 cycles [31] with a similar efficiency for the sorption of Pb ions sorption. However, the PVF/GO/Pb composite material has a porous structure showing a high conductivity. Moreover, the porous structure offers an enormous active surface area for the adsorption of electrochemically active materials and transition metal atoms. Additionally, ties between electrolyte water treatment membranes and ion-conducting polymers are made to prove that the underlying mechanisms that control ion sorption composites might also be employed to stimulate the development of new membrane materials for applications in fuel cells. This means that PVF/GO/Pb material will not be considered waste materials. This material displays a high intrinsic activity due to its crystallographic defects to catalyze hosts of electrochemical reactions, which are judged relevant in clean energy technologies.

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
