# Peer review of "Graphene Oxide/Polyvinyl Alcohol–Formaldehyde Composite Loaded by Pb Ions: Structure and Electrochemical Performance"

_polymers, 2022, doi:10.3390/polym14112303_

Round 1

Reviewer 1 Report

This research is about graphene oxide-polyvinyl alcohol formaldehyde composites. The research has its merits, yet there are a few issues that should be addressed before considering publication.

1. The research scope should be clearly stated. After reading the whole manuscript, I see that the authors mainly did two studies: using PVF-GO composite to remove Pb ions, and studying the properties and applications of PVF-GO-Pb composites.

2. These two topics are not that relevant to each other. Pb removal and water purification are more about environmental protection, while the dielectric and conductivity studies are more for the synthesis and application of new materials and devices. Hence the dielectric and conductivity studies should be a separate research paper with more detailed studies (using other metals instead of Pb, fabricating demo devices etc.) After all, Pb is toxic and should be replaced when possible.

3. Are the PVF-GO membranes reusable? How to desorb Pb from PVF-GO-Pb composites?

4. Characterizations of GO are missing.

5. How important is GO here? The GO content in the material is 0.2 wt%. The blank PVF without GO is missing. Can it adsorb any Pb? What is higher GO contents are used?

6. "The distribution of Pb ions  on the surface of the PVF/GO composite was investigated by the elemental mapping of the surface of the foam after the adsorption process of the Pb ions (Figure 6d)." Figure 6d is not the EDX mapping.

7. The language need to be carefully checked.

Author Response

This research is about graphene oxide-polyvinyl alcohol formaldehyde composites. The research has its merits, yet there are a few issues that should be addressed before considering publication.

We would like to thank the reviewer for the summary of our work. We are grateful for the time and energy you expended on our behalf.

  1. The research scope should be clearly stated. After reading the whole manuscript, I see that the authors mainly did two studies: using PVF-GO composite to remove Pb ions and studying the properties and applications of PVF-GO-Pb composites.

We appreciate the insightful suggestion concerning this issue. It is clearly stated in the last paragraph in the introduction section as: “The relation between the structure and the electrochemical behavior of the composite film related to the absorbed metal ions is not studied till now. Therefore, here dielectric spectroscopy was utilised to explore the segmental fluctuations of the prepared PVF/GO composite in comparison to the corresponding composites loaded with Pb ions. Additionally, the paper also intends to explore the effect of the influence of the Pb ions on the structural and morphological properties as well as the dielectric behavior including the electrical conductivity of the PVF/GO composites.”

  1. These two topics are not that relevant to each other. Pb removal and water purification are more about environmental protection, while the dielectric and conductivity studies are more for the synthesis and application of new materials and devices. Hence the dielectric and conductivity studies should be a separate research paper with more detailed studies (using other metals instead of Pb, fabricating demo devices etc.) After all, Pb is toxic and should be replaced when possible.

We agree with the reviewer. It is well known that the adsorbents could be regenerated for several times and after that they will be a west. Therefore, this research paper is focused on the structure and electrochemical properties of the composite film which act as an adsorbent to find new properties which may led to new applications. Therefore, here dielectric spectroscopy was utilised as a tool to explore the segmental fluctuations as molecular probe for structure of the created PVF/GO composite in comparison to the corresponding composites loaded with Pb ions. Additionally, the paper also intends to explore the effect the influence of the Pb ions on the structural and morphological properties as well as the dielectric behavior including the electrical conductivity of the PVF/GO composites.

The missing information was added in the introduction part.

  1. Are the PVF-GO membranes reusable? How to desorb Pb from PVF-GO-Pb composites?

Using 10% H2SO4 solution as mentioned in our previous work [Yosef, M; Fahmy, A; El Hotaby, W; Hassan, AM; Khalil, ASG; Anis, B. High performance graphene-based PVF foam for        lead removal from water. J. Mater. Res. Technol. 2020, 5, 11861–11875. https://doi.org/10.1016/j.jmrt.2020.08.011]

A comment is added to the manuscript.

  1. Characterizations of GO are missing.

The prepared GO was investigated in detail elsewhere [31]. Acorresponding comment is added to the manuscript to make this more clear for readers.

  1. How important is GO here? The GO content in the material is 0.2 wt%. The blank PVF without GO is missing. Can it adsorb any Pb? What is higher GO contents are used?

All these details were discussed elsewhere [31]. A coomment is added to the manuscript as ‘GO carries a high number of oxygen-containing functional groups, large surface area, well interconnected porous specifying plenty of oxygenated functional groups available for metal adsorption [] and it has further efficient antibacterial activity [].

  1. "The distribution of Pb ions on the surface of the PVF/GO composite was investigated by the elemental mapping of the surface of the foam after the adsorption process of the Pb ions (Figure 6d)." Figure 6d is not the EDX mapping.

We are grateful for the reviewer’s input. It was a mistake therefore, it was corrected.

  1. The language need to be carefully checked.

We would like to thank the reviewer for raising this issue. The language in the whole manuscript was revised and improved as much we can.

Reviewer 2 Report

The authors incorporated graphene oxide (GO) into PVF foam and investigated the electrochemical properties of the composite material with/without the adsorption of Pb ions. They used thorough morphological and spectroscopic characterization tools to elucidate the structure and chemistries of the system, with which to corroborate the conduction mechanism change that they proposed. Design of experiments is sound and the conductivity vs.  frequency & temperature plots in Figure 9 is a crucial an convincing piece of evidence to make the argument. Potential applications of the system are in the areas of energy storage (polymer electrolyte, capacitor etc.) and the work did shine light on the fundamental electrochemical processes. 

All the merits aside, the authors should revise the manuscript by correcting a handful of typographical errors before publishing. 

Author Response

The authors incorporated graphene oxide (GO) into PVF foam and investigated the electrochemical properties of the composite material with/without the adsorption of Pb ions. They used thorough morphological and spectroscopic characterization tools to elucidate the structure and chemistries of the system, with which to corroborate the conduction mechanism change that they proposed. Design of experiments is sound and the conductivity vs.  frequency & temperature plots in Figure 9 is a crucial an convincing piece of evidence to make the argument. Potential applications of the system are in the areas of energy storage (polymer electrolyte, capacitor etc.) and the work did shine light on the fundamental electrochemical processes. 

We thank the reviewer for the clear summary of our work. We are grateful for the time and energy you expended on our behalf.

All the merits aside, the authors should revise the manuscript by correcting a handful of typographical errors before publishing.

The quality of the whole manuscript was improved as you can see in this current draft.

Round 2

Reviewer 1 Report

Questions have been answered. References have been cited for supplementary information.